# Slaying (Yet Again) the Brain-Eating Zombie Called the “Isochore Theory”: A Segmentation Algorithm Used to “Confirm” the Existence of Isochores Creates “Isochores” Where None Exist

**DOI:** 10.3390/ijms23126558

**Published:** 2022-06-12

**Authors:** Dan Graur

**Affiliations:** Department of Biology & Biochemistry, University of Houston, Science & Research Building 2, 3455 Cullen Blvd., Suite #342, Houston, TX 77204-5001, USA; dgraur@uh.edu

**Keywords:** isochores, GC content, segmentation algorithms, benchmark simulations, isoSegmener, isoPlotter

## Abstract

The isochore theory, which was proposed more than 40 years ago, depicts the mammalian genome as a mosaic of long, homogeneous regions that are characterized by their guanine and cytosine (GC) content. The human genome, for instance, was claimed to consist of five compositionally distinct isochore families. The isochore theory, in all its reincarnations, has been repeatedly falsified in the literature, yet isochore proponents have persistently resurrected it by either redefining isochores or by proposing alternative means of testing the theory. Here, I deal with the latest attempt to salvage this seemingly immortal zombie—a sequence segmentation method called isoSegmenter, which was claimed to “identify” isochores while at the same time disregarding the main characteristic attribute of isochores—compositional homogeneity. I used a series of controlled, randomly generated simulated sequences as a benchmark to study the performance of isoSegmenter. The main advantage of using simulated sequences is that, unlike real data, the exact start and stop point of any isochore or homogeneous compositional domain is known. Based on three key performance metrics—sensitivity, precision, and Jaccard similarity index—isoSegmenter was found to be vastly inferior to isoPlotter, a segmentation algorithm with no user input. Moreover, isoSegmenter identified isochores where none exist and failed to identify compositionally homogeneous sequences that were shorter than 100−200 kb. Will this zillionth refutation of “isochores” ensure a final and permanent entombment of the isochore theory? This author is not holding his breath.

## 1. Introduction

In the 1970s, several authors discovered that, by fractionating mammalian genomic DNA and running it on either Cs_2_SO_4_−Ag+ or CsCl density gradients, a continuous distribution of DNA buoyant densities was observed, e.g., [1,2]. This distribution was wider and less symmetrical than that resulting from fractionating bacterial DNA. By using an arbitrary process of recurrent subsampling, followed by an equally arbitrary discretization of these continuous distribution curves, Cuny et al., proposed that genomic DNA is made of very large DNA segments of homogeneous GC content, which were termed “isochores,” and that these segments belong to a small number of isochore families characterized by identical or similar GC contents [3]. For example, according to [3], most of the human genome (93%) consists of four isochore families, whose GC contents are 36.7% GC (29% of the genome), 38.5% GC (33% of the genome), 42.9% GC (22% of the genome), and 49.2% GC (9% of the genome).

The isochore theory in all its reincarnations has been repeatedly falsified, e.g., [4,5,6,7,8,9,10,11]. Sadly, however, each refutation brought a new barrage of counter arguments by the proponents of the isochore theory and, like a classical brain-eating zombie, the theory refused to die. In response to the many falsifications, proponents and supporters of the isochore theory either redefined isochores, proposed alternative means of testing the theory, or both. For example, the number of isochore families in the human genome grew from four to five, the size of the isochores fluctuated widely, the GC contents of the isochore families periodically increased or decreased, and the adjective “homogenous” became a more nebulous “fairly homogenous.” Here, I deal with the latest iteration of the isochore theory, in particular, with the newest sequence segmentation methods that were put forward to salvage this zombie [12,13].

### 1.1. Isochores and the Human Genome Sequence

Many research papers published before genome sequencing became available used the GC content at third-codon positions (GC3) in protein-coding genes as a proxy for genomic GC content [14,15]. The main drawback of this method was that protein-coding regions comprise only ~1% of the entire human genome [16]. Moreover, codon usage biases can only be meaningfully assessed in four-fold degenerate sites (i.e., third-codon positions in which all possible point mutations result in synonymous substitutions). These sites, notwithstanding, constitute only ~17% of all coding regions. As could have been expected, GC3 failed to predict genomic GC content [5].

Many analyses of the first-draft sequence of the human genome [6] cast serious doubt on the existence of compositionally homogeneous isochores [8,9,11], while [6] concluded that “isochores” do not merit the prefix “iso.” The publication of the cow genome [7] added evidence against the validity of isochores.

### 1.2. Segmentation Algorithms

Cohen et al., analyzed the GC composition of the human genome [10] by using a segmentation algorithm which partitioned each genomic sequence into segments using a binary recursive method, as proposed by [17]. In the initial step of this procedure, a chromosome is segmented into two subsequences at a point that maximizes the difference in GC content between the adjacent subsequences. The process of segmentation was repeated recursively on all the subsegments, and the process was continued until the difference in GC content between two neighboring subsegments was no longer statistically significant. By using three criteria for defining isochores (distinctiveness, homogeneity, and a minimal length of 300 kb), Cohen et al., discovered that genomic segments that warrant the label ‘‘isochore’’ cover only 41% of the human genome and are nonuniformly scattered throughout the genome [10]. Moreover, only 4% of the homogeneous segments in the human genome could be labeled as isochores, and almost all such segments were GC-poor. Cohen et al., also found that a four-family model of putative isochores was the most parsimonious multi-Gaussian model that could be fitted to the empirical genomic sequence data [10], as opposed to the five-family isochore model [18]. These four Gaussians had mean GC contents of 35%, 38%, 41%, and 48%, which did not resemble the values previously found in the isochore literature. Finally, due to large overlaps between the families, it was impossible to classify genomic segments into isochore families reliably, according to compositional properties alone. These findings undermined the utility of the isochore theory and seemed to indicate that the isochore theory had reached the limits of its usefulness as a description of genomic compositional structures.

Cohen et al.’s [10] segmentation algorithm had one problem: it required user input. Specifically, the segmentation algorithm halted the segmentation process when the Jensen–Shannon divergence statistic [19], which measures the difference in GC content between both sides of the segmentation point, fell below a manually inputted threshold. All manual inputs are problematic as they can introduce experimenter bias, i.e., systematic errors that are attributable to a researcher’s preconceived beliefs, expectations, or desired results. isoPlotter replaced the manual input of a predetermined threshold with a dynamic threshold computed from the length and GC composition of the candidate subsequences, eliminating any need for user input [20,21]. Using isoPlotter, Elhaik and Graur found that homogeneous sequences longer than 300 kb covered less than 28% of total mammalian genomes and constituted less than 2% of all identified compositionally homogeneous domains [4]. 

As an alternative to Cohen et al.’s [10] algorithm and to isoPlotter, which eliminated all user input [20,21], Costantini et al. [12] suggested a new segmentation algorithm—later revised as isoSegmenter by Cozzi et al. [13]. These two algorithms are essentially all user input. Constantini et al.’s [12] method is as straightforward as it is deceitful. It is a perfect example of preconceived results obtained through the manipulation of methodology. In Costantini et al.’s [12] method, “chromosomal sequences of the finished human genome assembly were partitioned into non-overlapping 100-kb windows, and their GC levels were calculated.” This first step by itself assures that no homogenous segment shorter than 100,000 base pairs will be identified and that the internal compositional variation within each segment will be completely ignored. One example of a non-isochore that would be identified as an isochore by the algorithm in [12] is shown in Figure 1. In the next step, the GC levels of adjacent 100 kb windows in each chromosome were scanned “for jumps that were detectable on the basis of mean GC differences.” As a guideline, they focused on “jumps of at least 1–2% GC between adjacent candidate segments, although in rare cases smaller jumps were justified.” I note that the internal variation in GC content within each segment was again ignored. Thus, in many cases, adjacent segments that differed significantly in GC content were clustered together into large “isochores” and, conversely, adjacent segments that differed insignificantly in GC content were deemed to belong to different isochore families. Furthermore, as if these serious offences against the scientific method were not enough, Costantini et al. [12] allowed for exceptions in deciding that two adjacent isochores belonged to different isochore families through the vague statement that “in rare cases smaller jumps [than 1% GC] were justified”.

A “new-and-improved” algorithm by the isochore proponents, isoSegmenter, was put forward by Cozzi et al. [13]. isoSegmenter goes a step further than [12] by making sure that exactly five isochore families will be identified whether they exist or not. In contrast to [12], in which the number of isochore families is not predetermined, isoSegmenter leaves nothing to chance. isoSegmenter partitions the genome into non-overlapping 100 kb-length windows and does not pay attention to the compositional variation within each segment. Next, the GC content is computed for each window and, based on the GC content, the window is assigned to one of five predetermined isochore families, as dictated by [18]: L1 (<37% GC), L2 (37–41% GC), H1 (41–46% GC), H2 (46–53% GC), and H3 (>53% GC). Finally, neighboring windows belonging to the same family are merged to form long, continuous segments. IsoSegmenter is one of the most blatant examples of “garbage in, garbage out” methodology. Thus, a window with a GC frequency of 52.999% is deemed to be an H2 isochore, while one with a GC frequency of 53.001% is said to be an H3 isochore. By ignoring the variation within each 100 kb segment, and by throwing into the mixture five colors—one for each isochore family, one can obtain a very decorative representation of wishful genomic thinking, full of sound and fury, signifying nothing. In principle, there is no need to test and compare isoSegmenter with isoPlotter, as isoSegmenter is clearly the product of a self-delusional paradigm. However, the author of this paper believes in giving each method an equal opportunity to fail.

## 2. Results

One-hundred simulations were generated for each category of equal-length and variable-length domain sequences, for a total of 1100 equal-domain-length and 900 variable-domain-length sequences. The simulated sequences generated were segmented using isoPlotter and isoSegmenter, and the predictions of each algorithm were compared to the ground truth and scored. A summary of the overall performance metrics of both algorithms is shown in Table 1.

I note that the medians of all three performance metrics for isoSegmenter were much closer to zero than to the mean, implying that all the performance metrics for isoSegmenter were skewed upwards by high-performance outliers. The medians for isoPlotter, on the other hand, were much closer to the mean, indicating more consistent performance. Because of this skew in isoSegmenter, the Wilcoxon test was used instead of the standard two-mean *t*-test. Overall, isoPlotter had a 10–20-fold higher median sensitivity, precision rate, and Jaccard index compared to isoSegmenter, and the differences in these performance metrics were all highly statistically significant.

Let us now examine how domain size influences the performance of isoPlotter and isoSegmenter. With very large domain sizes (above 300 kb), isoSegmenter exhibited high performance metrics (Figure 2). As the size of the domains decreased, however, all performance metrics rapidly and precipitously declined to zero. In contrast, isoPlotter displayed consistent performance levels across all sequences tested (Figure 3). For isoPlotter, sensitivity remained between 50% and 60% across all types of equal and variable domain sequences. However, isoPlotter’s precision rate and Jaccard index increased as the number of domains increased (especially with equal-length domains), and both these indices were higher among equal-length domains than in variable-length ones.

To better understand the performance changes across domain lengths, the average predicted domain lengths for both isoPlotter and isoSegmenter on the equal-length datasets were plotted against the true domain sizes (Figure 4). Ideally, the predicted lengths would exactly match the ground truth, leading all data points to lie on a 45° diagonal through the origin. Deviations of data points away from this reference line reflected prediction inaccuracies. From this figure, it is apparent that the average prediction length of isoSegmenter did not decrease below 200,000 base pairs (bp), even while segmenting sequences with domain lengths as small as 10,000 bp. In contrast, isoPlotter tended to slightly oversegment sequences into smaller domains, but its predictions were closer to the reference 45° diagonal. Moreover, unlike isoSegmenter, isoPlotter’s average prediction lengths were correlated with the true domain sizes.

## 3. Discussion

In this study, I used a series of controlled, randomly generated simulated sequences to conduct a benchmark comparison of isoPlotter and isoSegmenter, two compositional segmentation algorithms that deliver highly contradictory predictions. The main advantage of using simulated sequences is that, unlike real data, the exact start and stop point of the isochore or homogeneous compositional domain is predetermined. This enables me to directly compare the domain predictions of the two algorithms.

The performance of isoSegmenter against the benchmark sequences was dismal. In comparison, isoPlotter exhibited a much higher performance because of its ability to deal with a wider range of domain lengths. It did, however, exhibit a systematic tendency to predict domain sizes that were slightly smaller than the ground truth.

Cozzi et al., criticized algorithms such as isoPlotter as having “no biological relevance” [13]. However, the most important priority of any segmentation algorithm is to make predictions that accurately reflect the underlying true genomic sequence. A segmentation algorithm that superimposes unrepresentative, artificial boundaries onto a genomic sequence with no grounds in the true compositional profile is one which truly lacks biological relevance. isoSegmenter is an algorithmic exercise in obtaining the desired results at all costs without taking into consideration the true genomic data. 

These systematic failures of isoSegmenter raise questions regarding the validity of the findings of [13], as well as the validity of multiple other papers that used isoSegmenter as a key part of their computational analyses, e.g., [18,22,23,24,25,26,27,28,29].

Interestingly, this algorithm has an inbuilt self-delusionary routine that discovers “isochores” in random sequences, in artificial sequences, and in literary masterpieces. For example, isoSegmenter and its parent algorithm [12] were used to identify isochores in the *Drosophila* genome [23], which according to the proponents of the isochore theory is not supposed to have isochores. 

As an exercise in satire, I translated Herman Melville’s 1851 novel *Moby-Dick* into DNA by using the method in [30] and applied isoSegmenter to the resulting sequence. Unsurprisingly, we found that this famous great whale of a novel is made of “isochores.”

Will the isochore theory be finally put to rest? We are not optimistic; brain-eating zombies are notoriously hard to kill.

## 4. Materials and Methods

### 4.1. Simulated Genomic Sequences

For my first analysis, I generated 1 Mb-long DNA sequences. Each sequence was divided into equal-length segments. Several divisions were used: 1, 2, 4, 5, 10, 20, 25, 40, 50, 80, and 100. If, for example, a simulated sequence was divided into 10 domains, it was composed of ten 100-kb equal-length domains. Each domain was randomly assigned to one of the five isochore families (L1, L2, H1, H2, and H3) with GC contents of 22.8%, 33.2%, 22.7%, 11.2%, and 3.01% [31]. The precise GC sequence of each domain was randomly selected from the GC content range of the assigned family. Adjacent domains were always assigned to a different isochore family. Simulated sequences were converted into the FASTA format using Biopython [32].

For the second analysis, a more realistic segmentation method was used. This segmentation method partitioned the initial genomic sequence into domains of varying lengths according to a power-law distribution with an exponent of α = −2.55, as stipulated in [10]. For this analysis, we generated simulated genomic sequences, each with a length of 5 Mb. The number of partitions was 2, 4, 6, 8, 10, 20, 30, 40, and 100. Using the transformation method described by [33], the lengths of each simulated domain were randomly sampled from a power-law distribution with *x*min = 10,000 bp and α = −2.55 using the inverse cumulative distribution function. Afterwards, the domain lengths were normalized so that their total sequence length was 5 Mb.

Examples of GC profiles for equal-length and variable-length simulated segments are shown in Figure 5.

### 4.2. Scoring of Predicted Domains

Both isoPlotter and isoSegmenter were run on each of the simulated sequences. Following Elhaik et al. (2010), a predicted domain was deemed a true positive (TP) if both its boundaries matched the actual boundary positions of the domain within ± 5% of the predicted domain size. A false positive (FP) was a predicted domain for which one or both predicted boundaries did not match the ground truth boundaries. A false negative (FN) domain was a ground-truth domain that was not successfully predicted by the algorithm (Figure 6).

Three key performance metrics, i.e., sensitivity, positive predictive value (or precision rate), and the Jaccard index, were compared side by side on identical benchmark simulations for isoPlotter and isoSegmenter as follows:Sensitivity = TP/(TP + FN)
Precision = TP/(TP + FP)
Jaccard Index = TP/(TP + FP + FN)
where TP, FP, and FN stand for true positive, false positive, and false negative results, respectively.

## Figures and Tables

**Figure 1 ijms-23-06558-f001:**
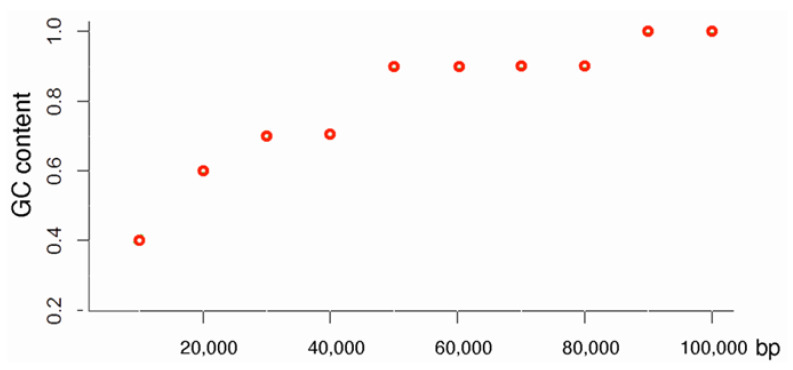
Identifying an “isochore” where none exists by the algorithms of Cozzi et al., and Costantini et al. [12,13]. According to both algorithms, chromosomal sequences are first “partitioned into non-overlapping 100-kb windows.” In the next step, the GC level is calculated as the mean GC content within the “window.” Let us assume that the sequence above represents one such window, and that the GC contents within its ten 10 kb nonoverlapping sub-windows vary, as shown. The variation will be ignored by the algorithms and the entire 100 kb window will be deemed to be an “isochore” with a mean GC content of 0.8, even though none of the sub-windows has such a GC content.

**Figure 2 ijms-23-06558-f002:**
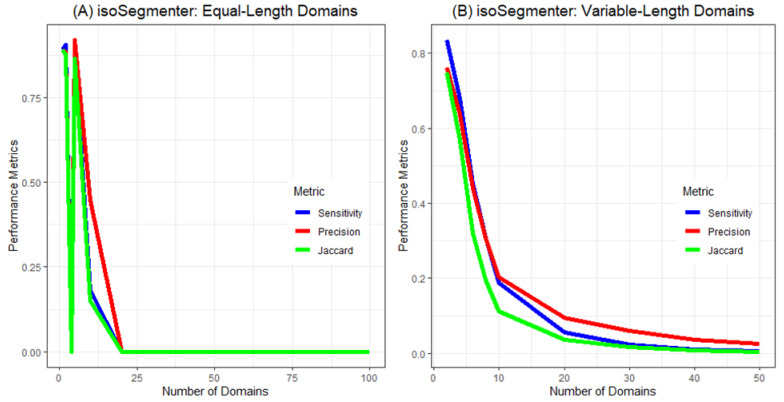
Performance metrics for isoSegmenter across numbers of equal-length domains (**A**) and variable-length domains (**B**).

**Figure 3 ijms-23-06558-f003:**
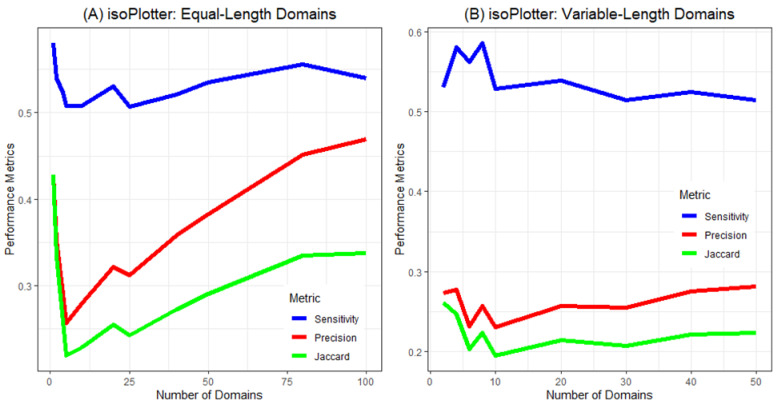
Performance metrics for isoPlotter across numbers of equal-length domains (**A**) and variable-length domains (**B**).

**Figure 4 ijms-23-06558-f004:**
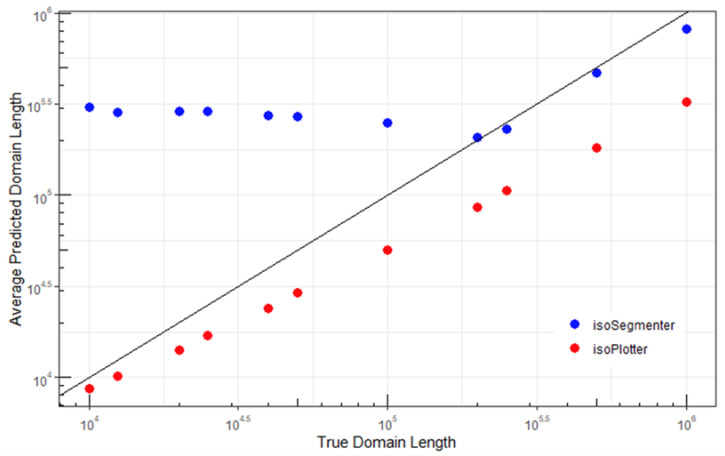
Comparison of average predicted domain lengths for isoPlotter and isoSegmenter to the actual domain lengths in the equal-domain-length simulated sequences.

**Figure 5 ijms-23-06558-f005:**
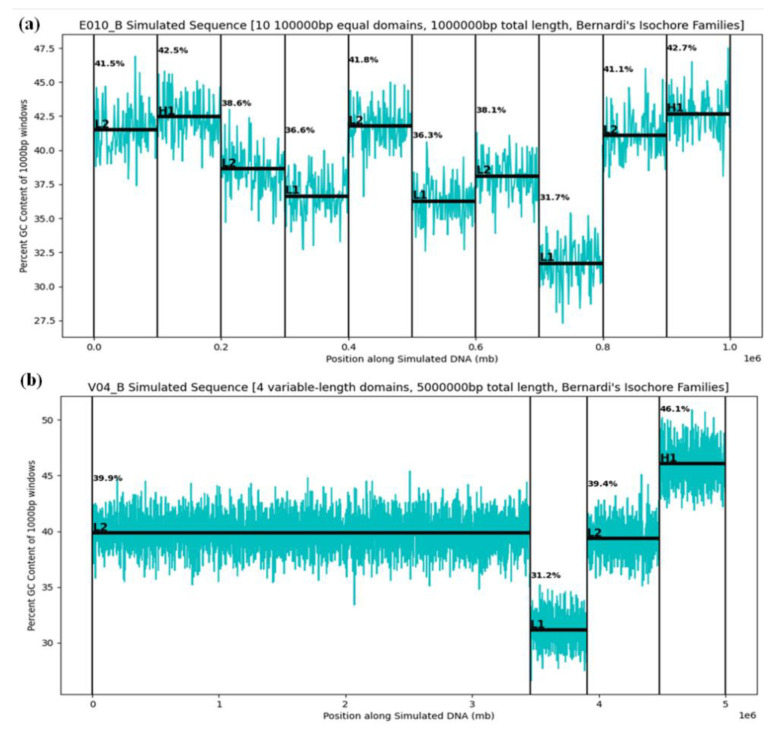
The GC content profiles of (**a**) a 1 Mb simulated sequence with ten equal-length domains and (**b**) a 5 Mb simulated sequence with four variable-length domains. Isochore family and mean GC percentage are indicated for each simulated domain.

**Figure 6 ijms-23-06558-f006:**
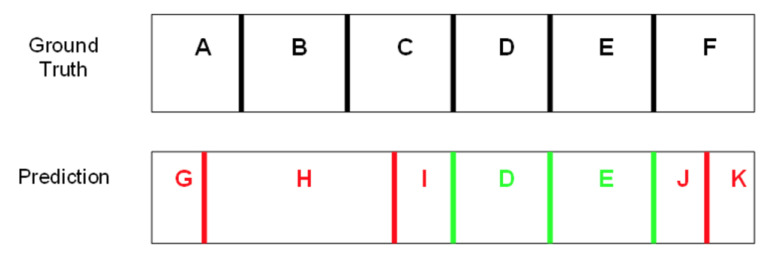
An example of how domain predictions are scored. Green boundaries indicate correct predictions. Domains marked with green letters indicate correctly predicted domains. Red boundaries indicate incorrect predictions. Domains marked with red letters indicate incorrectly predicted domains. A domain is only considered to be correctly predicted if both boundaries align with the ground truth. In this example, the true positives (TP) are D and E; the false positives (FP) are G, H, I, J, and K; and the false negatives (FN) are A, B, C, and F.

**Table 1 ijms-23-06558-t001:** Performance metrics of isoPlotter and isoSegmenter on the simulated sequence dataset.

	Mean for isoPlotter	Median for isoPlottter	Mean for isoSegmenter	Median for isoSegmenter	*p*-Value *
Sensitivity	0.5362	0.52	0.2714	0.025	*p* < 10^−16^
Precision	0.3122	0.2903	0.2854	0.0714	*p* < 10^−16^
Jaccard Index	0.26	0.2321	0.2394	0.0196	*p* < 10^−16^

* *p*-values were computed using a two-sample Wilcoxon rank sum test, under the null hypothesis of equal medians for isoPlotter and isoSegmenter.

## Data Availability

The coding files were deposited in https://github.com/RackS103/Segmentation-Algorithm-Comparison (accessed on 1 June 2022).

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
