# Peer review of "Slaying (Yet Again) the Brain-Eating Zombie Called the “Isochore Theory”: A Segmentation Algorithm Used to “Confirm” the Existence of Isochores Creates “Isochores” Where None Exist"

_ijms, 2022, doi:10.3390/ijms23126558_

Round 1
Reviewer 1 Report
Upon first reading of the title of this manuscript it seemed as though
the method identifies confirms the existence of isochores. As second
reading of the title clarified that that is not the case. However,
i would prefer a title that doesn't include "isochores ... exist",
but rather that the segmentation algorithm identifies false isochores.
More problematic is the choice of the +/- 5%. If one chooses
instead 0.0001% then hopefully all instances will be false and the
method will look even more horrible. If one chooses 100% then many
more instances will be positive and the algorithm will look much
better. What is the correct percentage to choose? What is the
effect of the percentage? Is there a better way to measure accuracy
without an arbitrary cutoff.
Minor comments: On page 8 you define 'positive predictive value' but
you call it 'precision' elsewhere in the figures, tables and text. Yes,
they are the same thing, but then why not label all the sensitivities as
'recall', also the same thing.
On page 9 the figure is labelled as 'Figure 2' but it is not the
second figure. Indeed the other figures are not labelled at all.
On page 3 "Using isoPlotter, Elhaik et Graur found" -> "Using
isoPlotter, Elhaik _and_ Graur (2014) found"
On page 3, "an isochore is shown in Figure In the next step" is not
a sentence.
On page 4, "With very large domain sizes (above 300kb), isoSegmenter
exhibted high performance metrics" needs a reference to Table ? /
Figure ?.
On page 5, fourth line from the bottom. The reference to literary
masterpieces is mentioned before it is introduced on the next page.
On page 6, "... 20,30,40, and Using" needs to be rewritten.
Author Response
Upon first reading of the title of this manuscript it seemed as though
the method identifies confirms the existence of isochores. As second
reading of the title clarified that that is not the case. However,
i would prefer a title that doesn't include "isochores ... exist",
but rather that the segmentation algorithm identifies false isochores.
Hope the new title is clearer:
Slaying (Yet Again) the Brain-Eating Zombie Called the “Isochore Theory”: A Segmentation Algorithm Used to “Confirm” the Existence of Isochores Creates “Isochores” Where None Exist
Please do not allow hyphenation of words at the end of lines.
More problematic is the choice of the +/- 5%. If one chooses
instead 0.0001% then hopefully all instances will be false and the
method will look even more horrible. If one chooses 100% then many
more instances will be positive and the algorithm will look much
better. What is the correct percentage to choose? What is the
effect of the percentage? Is there a better way to measure accuracy
without an arbitrary cutoff.
Since we have used the same criteria for both isoSegmenter and isoPlotter, the 5% cuttoff doesn’t matter much.
On page 8 you define 'positive predictive value' but
you call it 'precision' elsewhere in the figures, tables and text. Yes,
they are the same thing, but then why not label all the sensitivities as
'recall', also the same thing.
Fixed
On page 9 the figure is labelled as 'Figure 2' but it is not the
second figure. Indeed the other figures are not labelled at all.
Fixed all figure numbers.
On page 3 "Using isoPlotter, Elhaik et Graur found" -> "Using
isoPlotter, Elhaik _and_ Graur (2014) found"
corrected
On page 3, "an isochore is shown in Figure In the next step" is not
a sentence.
A figure number and a period was missing. Fixed
On page 4, "With very large domain sizes (above 300kb), isoSegmenter
exhibted high performance metrics" needs a reference to Table ? /
Figure ?.
Apologies for the mix-up with the figure numbers. Fixed.
On page 5, fourth line from the bottom. The reference to literary
masterpieces is mentioned before it is introduced on the next page.
Fixed
On page 6, "... 20,30,40, and Using" needs to be rewritten.
Fixed
Reviewer 2 Report
General comments
The manuscript by Dan Graur focuses on the isochore theory, which has been proposed more than 40 yrs ago. Over the years, particularly since the publication of the human genome (2001) and of the bovine genome (2009), the theory has been repeatedly falsified.
The article is clearly written and covers latest literature about isochores. Graur falsifies the theory again and he demonstrates that the popular software called isoSegmenter is not able to find any meaningful isochore structures in simulated data where true segment boundaries are known. Furthermore, he compares isoSegmenter with isoPlotter, a technique previously criticized by the authors of the isoSegmenter.
Minor comments
I found some minor issues which should be corrected before acceptance of the article.
Subsections are wrongly numbered, e.g., subsection ‘1.’ occurs twice on page 2. Furthermore, in Material and Methods, there is a subsection ‘4.’ without ‘1.-3.’
The numbering of figures in the figure legends is missing.
[p2] There is a reference mentioned (Bernardi et al. 1985) which might be related to reference [18] in the reference list but this is not clear.
[p3l4] The reference to ‘Elhaik et Graur’ should read ‘Elhaik and Graur’.
[p4] The third paragraph in the result section could be strengthen by offering the read a figure (as supplementary figure).
[p6] I quite like the analysis of Melville’s novel. Maybe, the author can show the results (figure?) in the supplement as well.
[p6] In the second paragraph of the M&M I found ‘Using’ instead of ‘using’.
[Appendix] Figures are not referenced in the main text.
Author Response
Subsections are wrongly numbered, e.g., subsection ‘1.’ occurs twice on page 2. Furthermore, in Material and Methods, there is a subsection ‘4.’ without ‘1.-3.’
Fixed
The numbering of figures in the figure legends is missing.
Fixed
[p2] There is a reference mentioned (Bernardi et al. 1985) which might be related to reference [18] in the reference list but this is not clear.
Fixed
[p3l4] The reference to ‘Elhaik et Graur’ should read ‘Elhaik and Graur’.
Fixed
[p4] The third paragraph in the result section could be strengthen by offering the read a figure (as supplementary figure).
Fixed
[p6] I quite like the analysis of Melville’s novel. Maybe, the author can show the results (figure?) in the supplement as well.
I am writing an article on apophenia (i.e., the tendency to perceive a connection or meaningful pattern between unrelated or random things) in molecular evolution. The “isochores” of Moby-Dick will be one example.
[p6] In the second paragraph of the M&M I found ‘Using’ instead of ‘using’.
Fixed
[Appendix] Figures are not referenced in the main text.
Fixed
